# Chronic disaster impact: the long-term psychological and physical health consequences of housing damage due to induced earthquakes

Katherine Stroebe [iD],[1] Babet Kanis,[1] Justin Richardson,[1] Frans Oldersma,[2] Jan Broer,[3] Frans Greven,[4] Tom Postmes[1]

► Prepublication history and supplemental material for this paper is available online. To view these files, please visit the journal online (http://dx.doi.org/10.1136/bmjopen-2020-040710).

[1]Department of Social Psychology, Rijksuniversiteit Groningen, Groningen, The Netherlands
[2]Department for Statistics and Research, Municipality of Groningen, Groningen, The Netherlands
[3]ABPG, Municipal Health Services, Groningen, The Netherlands
[4]Department of Environmental Health, Municipal Health Services, Groningen, The Netherlands

**Correspondence to**
Dr Katherine Stroebe;
k.e.stroebe@rug.nl

## ABSTRACT

**Objectives** To evaluate the long-term (psychosomatic) health consequences of man-made earthquakes compared with a non-exposure control group. Exposure was hypothesised to have an increasingly negative impact on health outcomes over time.

**Setting** Large-scale gas extraction in the Netherlands causing earthquakes and considerable damage.

**Participants** A representative sample of inhabitants randomly selected from municipal population records; contacted 5 times during 21 months (T1: N=3934; T5: N=2150; mean age: 56.54; 50% men; at T5, N=846 (39.3%) had no, 459 (21.3%) once and 736 (34.2%) repeated damages).

**Main measures** (Psychosomatic) health outcomes: self-rated health and Mental Health Inventory (both: validated; Short Form Health Survey); stress related health symptoms (shortened version of previously validated symptoms list). Independent variable: exposure to the consequences of earthquakes assessed via physical (peak ground acceleration) and personal exposure (damage to housing: none, once, repeated).

**Results** Exposure to induced earthquakes has negative health consequences especially for those whose homes were damaged repeatedly. Compared with a no-damage control group, repeated damage was associated with lower self-rated health (OR:1.64), mental health (OR:1.83) and more stress-related health symptoms (OR:2.52). Effects increased over time: in terms of relative risk, by T5, those whose homes had repeated damage were respectively 1.60 and 2.11 times more likely to report poor health and negative mental health and 2.84 times more at risk of elevated stress related health symptoms. Results for physical exposure were comparable.

**Conclusion** This is the first study to provide evidence that induced earthquakes can have negative health consequences for inhabitants over time. It identifies the subpopulation particularly at risk: people with repeated damages who have experienced many earthquakes. Findings can have important implications for the prevention of negative health consequences of induced earthquakes.

## INTRODUCTION

Recent years have seen a rise in induced seismicity due to human activities such as

### Strengths and limitations of this study

► The present study employs a longitudinal panel design with five measurement points to study (pschosomatic) health consequences of man-made earthquakes caused by gas extraction.

► The study has an exposed (residents with damage to housing) and a non-exposed (residents with no damage) control group.

► Two health measures (self-rated health; Mental Health Inventory) were previously validated, and the third was an adaptation of a previously validated symptoms list.

► Younger respondents were somewhat under-represented in our sample.

► There was 45.3% attrition over time but attrition was no different for the exposed versus non-exposed groups and was unrelated to health outcomes.

fracking, mining or gas extraction. This development is expected to continue. While smaller in magnitude than natural seismicity, induced seismicity can expose populations to considerable physical (eg, damage to housing) and social risks (eg, conflicts between residents and institutions). Moreover, this exposure is recurrent and chronic over time. While there is some insight into the long-term health risks of naturally occurring seismicity, little is known about the impact of induced seismicity. Given the increased use of energy technologies associated with seismicity, also in densely populated areas, knowledge of its health impact is important[1 2] (see also table 1 for definitions of gas-extraction related terminology).

Naturally occurring seismicity is associated with mental health problems in survivors (eg, depression, post-traumatic stress disorder (PTSD)).[3–5] While (some) more studies have been considering the longitudinal health effects of seismicity, lack of

**Table 1** List of definitions

| | |
|---|---|
| Conventional gas extraction | Extraction through drilling in deep subsoil reservoirs without the injection of chemical liquids. |
| Fracking | A stimulation technique in which a rock is fractured by a pressured liquid in order to extract oil or gas from wells. |
| Induced seismicity | Seismic events that are a result of human activity. |
| Natural seismicity | Seismic events that have a natural cause (eg, volcanic eruption). |
| Peak ground acceleration | Measure of the maximum increase in ground motion during an earthquake, recorded by a ground motion sensor. |
| Psychosomatic health | Health outcomes involving both mind and body. |
| Richter scale | Measure of strength of earthquakes with a logarithmic scale. |
| Shale gas | A natural gas that is trapped in fine grained sediment in rock. |
| Unconventional gas extraction | Gas reservoirs that require a special stimulation technique to extract gas (eg, by injecting large quantities of fluids underground). |

longitudinal design and an unexposed control group have been highlighted as major concerns for studies of natural disasters.[3 5 6] Moreover, the impact of natural seismicity cannot be equated with that of induced seismicity for several reasons: systematic reviews suggest that there is *lower* prevalence of mental health impairment for natural compared with human/technological disasters[7 8] (but see Ref. 9). Additionally, different stressors are at play: natural seismicity can be of greater magnitude, causing death and extensive damages to buildings. For induced seismicity, the maximum magnitude of earthquakes tends to be smaller.[10 11] Risks involve damage to property and an incremental impact on health, as residents are exposed to long-term stressors (eg, damages; changing community relations; conflicts of interest with powerful institutions).[12 13]

Factual information regarding the health impact of induced seismicity is sparse. Cross-sectional self-report studies[14–16] and an evaluation of health records of exposed adults[17] in the context of unconventional gas extraction suggest associations between induced seismicity and increased (psychosomatic) health symptoms (eg, sleep disruption, headaches, stress). It is difficult to draw conclusions regarding the impact of seismicity from such studies: exposure to (the consequences of) seismicity is not distinguished from other risk factors (eg, wastewater injections). Additionally, most studies lack a non-exposed control group and a reliable baseline, and we know of none that consider the longitudinal effects of exposure.

This lack of information regarding the (long-term) impact of induced seismicity on health is problematic.

The occurrence of induced earthquakes is increasingly common across the globe: 1174 projects worldwide report induced seismicity.[18] High-profile cases of induced earthquakes have occurred in Oklahoma, USA, and (on a smaller scale) Lancashire, UK.[19 20] There are rising concerns regarding the consequences thereof within exposed populations, coupled with calls to policy makers for monitoring and contingency planning.[19] Policy makers need to weigh the wider economic benefits against potential drawbacks for exposed residents.[21]

The present work was designed to address the lack of information regarding the long-term impact of induced seismicity for residents: it studies the longitudinal (psychosomatic) health impact of induced seismicity on a group exposed to the consequences of seismicity (damage to housing) versus a control group not exposed to these consequences. The study was conducted in the province of Groningen, Netherlands, where conventional gas extraction from the largest gas field in Europe takes place (see online supplemental figure S1 for more information on seismicity in this province). While the magnitude of seismic events (up to 3.6 Richter) is generally considered 'light', their magnitude has increased over the past 30 years, making this a slow-onset disaster, and their impact is felt well beyond the gas field boundaries. The recurrent earthquakes damage housing in a region not prepared for seismic activity[22] and the governmental response to damage compensation has been considered inadequate.[23]

The present study is novel in charting the chronic impact of exposure to damage on health over a time period of almost 2 years, on a large sample. We tested the following hypotheses: (1) exposure versus non-exposure will have a negative impact on (psychosomatic) health outcomes. (2) increases in exposure are related to poorer health outcomes.

## METHOD
### Setting and exposure
The study was conducted in the province of Groningen, Netherlands, where conventional gas extraction from the largest gas field in Europe takes place. Exposed residents experience rising concerns about physical safety, loss of property value and uncertainty about the future.[23 24] The benefits of extraction flow to the operator (the Netherlands petroleum company, NAM) and the national government, while damage repair and compensation by these entities has been criticised as being inadequate.[23]

Seismicity has increased over time. While the magnitude of seismic events (up to 3.6 Richter) is generally considered 'light', their impact is felt well beyond the gas field boundaries. Also, multiple factors (limited depth and high rates of occurrence of earthquakes; surface constitution) contribute to considerable damage to housing in a region not prepared for seismic activity.[22] For these reasons, documented damage has proven the most proximal measure of exposure, compared with indicators of seismicity.[25]

## Sample and recruitment

A stratified random sample was drawn of 25000 residents of the province of Groningen, aged 16 and over, from the official municipal population records which is a complete register of all legal residents. Sampling occurred in areas where damage is reported and from outlying areas where this is not the case. Postal-code areas that were rural and strongly affected by damage were oversampled. (In the Netherlands, four-number postal-code areas provide reasonably accurate geographic positioning, while preserving anonymity. Data about damage in each area were provided by the institution handling damage claims, the Centrum voor Veilig Wonen.) Residents received letters with personal login codes and one reminder. Eighteen per cent (n=4577) signed up for the study, and later received invitations to all questionnaires. Of these 4577, 86% (3934) filled out the first questionnaire. Baseline equivalence of non-exposed and exposed groups was assessed. Differences between groups were significant but small. Those with multiple damage to homes were slightly younger ($r^2$=0.014), more highly educated (Cramer's V=0.062) and more likely to be men (V=0.072). The first two characteristics suggest the exposed group might be slightly healthier. We statistically controlled for these characteristics.

## Data sources
### Procedure

Questionnaires were sent via an email link or by post. A reminder was sent after 2 weeks. Participants (T1: N=3934; T5: N=2150) completed measures at 5 time points during 2 years (T1: February 2016, T2: June 2016; T3: November 2016; T4: April 2017; T5: November 2017; see table 2).

## Study variables

*Exposure to consequences of gas extraction* was operationalised in two ways. Chronic physical exposure to ground motion was assessed by the cumulative peak ground acceleration ($PGA_{cum}$) on the basis of 'shakemaps' provided by the Dutch geological survey (KNMI). (KNMI calculates shakemaps based on motion sensor readings. For each participant, the PGA of all events modelled by KNMI between 2012 and 2017 was summed, to create an index of exposure to ground motion before and during the study.) Personal exposure to damage due to ground motion was assessed at every timepoint by asking participants to indicate how often their home had been damaged (never, once or multiple times) (see online supplemental table S1 for demographic characteristics by level of damage exposure).

*Demographic variables* included gender, age and completed education level (categorised into 'low' (no, elementary or prevocational education), 'middle' (secondary or vocational education) or 'high' (higher education) level of education).

*(Psychosomatic) health outcomes* were assessed at (almost) all time points (table 2) as follows via:
1. The WHO and Statistics Netherlands recommended validated health survey item assessing *self-rated health*[26] ('how good is your health in general?', from 'very poor' to 'excellent' on a 5-point scale), which is part of the Short Form Health Survey (SF-36).[27]
2. *Stress-related health* symptoms were based on a validated scale of symptoms of disaster impact.[28] This list of symp-

**Table 2** Demographic characteristics of participants participating in separate measurements: total number of participants participating in that measurement, decline of number of participants participating as compared with the number of participants participating at T1, mean age, distribution of level of education, distribution of personal exposure to damage due to gas extraction, distribution of gender and amount of participants that completed the three health measures in that measurement. Netherlands 2016–2017.

| | | T1 February 2016 | T2 June 2016 | T3 November 2016 | T4 April 2017 | T5 November 2017 |
|---|---|---|---|---|---|---|
| Total N | | 3934 | 3153 | 2638 | 2351 | 2150 |
| Attrition (compared with T1) | | – | 19.9% | 32.9% | 40.2% | 45.3% |
| Age (mean) | | 56.54 | 57.74 | 57.72 | 58.90 | 59.98 |
| Level of education (N) | Low | 968 (24.6%) | 772 (24.5%) | 616 (23.4%) | 589 (25.1%) | 535 (24.9%) |
| | Middle | 1252 (31.8%) | 970 (30.8%) | 815 (30.9%) | 713 (30.3%) | 639 (29.7%) |
| | High | 1533 (39.0%) | 1238 (39.3%) | 1068 (40.5%) | 944 (40.2%) | 852 (39.6%) |
| Gender (N) | Male | 1967 (50.0%) | 1547 (49.1%) | 1306 (49.5%) | 1182 (50.3%) | 1068 (49.7%) |
| | Female | 1849 (47.0%) | 1480 (46.9%) | 1231 (46.7%) | 1097 (46.7%) | 990 (46.0%) |
| Exposure to damage (N) | None | 1477 (37.5%) | 1204 (38.2%) | 1027 (38.9%) | 910 (38.7%) | 846 (39.3%) |
| | One time | 913 (23.2%) | 626 (19.9%) | 554 (21.0%) | 505 (21.5%) | 459 (21.3%) |
| | Multiple | 1057 (26.9%) | 1055 (33.5%) | 940 (35.6%) | 775 (33.0%) | 736 (34.2%) |
| Perceived health (N) | | 3821 (97.1%) | – | 2540 (96.3%) | 2206 (93.8%) | 2059 (95.8%) |
| Stress-related health symptoms (N) | | 3767 (95.8%) | – | 2533 (96.0%) | 2206 (93.8%) | 2045 (95.1%) |
| Mental health (N) | | 3711 (94.3%) | 2819 (89.4%) | 2501 (94.8%) | 2179 (92.7%) | 2021 (94.0%) |

toms was shortened by authors (JB, FG, TP): symptoms associated with chronic stress were retained (notably, at the level of individuals who suffer these complaints, they are referred to as 'medically unexplained' because they can have multiple sources, among which is chronic stress). Consequences of exposure to toxic substances and noise (eg, hearing problems) were deemed irrelevant for earthquakes and removed. Ten symptoms (stomach problems, heart palpitations, headaches, dizziness/lightheadedness, sensitivity to light/sounds, muscle/joint pains, irritability, memory/concentration problems, insomnia, tiredness) were assessed by asking 'how often have you experienced the following complaint(s) in the past 4 weeks' with response options 'never, rarely, occasionally, often, most times, continuously'. Aggregate health index scores were computed for stress-related health symptoms, so that individuals have a score of 0 to 100, with 100 representing optimal health. Psychometric properties of the aggregate scale were adequate. Correlations among items ranged from ordinal rho 0.26 to 0.72 (median=0.39). A single factor explained 46% of variance. Scale reliability was good with omega=0.90.

3. The five-item validated Mental Health Inventory (MHI-5), part of the SF-36, measuring general *mental health*.[27 29] The MHI-5 has a score of 0 to 100. A score of 100 represents optimal mental health.

### Data management and analysis

Analyses controlled for age, gender and education level. Analyses were weighted to correct for sampling effects of age, gender and degree of exposure of postal-code areas. (As mentioned, we oversampled rural areas as well as the most heavily exposed areas. The geographical weighting was added to control for this overrepresentation.) The weights were developed to counteract any potential distortive effect due to age composition, among others (eg, because younger people were under-represented, see Results section). We report the weighted results. The unweighted results were very similar.

To assess the impact of exposure to gas extraction on health over time, we constructed multilevel conditional growth models on the three health indices with damage to housing as the (between group) predictor.[30] Participants with missing data on the health indices were retained, as multilevel modelling is robust to missingness in estimation of model outcomes.

Models were tested in a stepwise approach, first including control variables (gender, age, level of education) and time. At the next step, physical exposure ($PGA_{cum}$) was added, followed by earthquake damage at time 1 and the increase of damage since time 1. The final model included the interaction between damage and time. Model fit was compared to assess which variables best predicted health outcomes. The best fitting models were those including the interaction of damage by time (see table 3).

**Table 3** Results of multilevel conditional growth models: unstandardised parameter estimates and SEs for the association between time, damage and the interaction between time and damage on perceived health, stress-related health symptoms and mental health—adjusted for gender, age, level of education and ground motion (cumulative PGA). Netherlands 2016–2017.

| | Perceived health | Stress-related health symptoms† | Mental health |
|---|---|---|---|
| Gender | −0.05* | −5.40*** | −2.68*** |
| | (0.02) | (0.49) | (0.46) |
| Age | −0.01*** | −0.02 | 0.07*** |
| | (0.001) | (0.02) | (0.02) |
| Level of education (middle) | 0.08* | 0.61 | 1.01 |
| | (0.03) | (0.67) | (0.62) |
| Level of education (high) | 0.24*** | 3.02*** | 2.94*** |
| | (0.03) | (0.63) | (0.59) |
| Cumulative PGA | −0.001 | 0.03 | −0.01 |
| | (0.004) | (0.09) | (0.08) |
| Time | −0.01 | −0.25* | −0.49*** |
| | (0.01) | (0.13) | (0.15) |
| Damage (one time) | −0.01 | −0.46 | −0.27 |
| | (0.03) | (0.75) | (0.63) |
| Damage (multiple) | −0.12*** | −4.31*** | −3.35*** |
| | (0.03) | (0.76) | (0.65) |
| Time * damage (one time) | −0.02 | −0.13 | −0.07 |
| | (0.01) | (0.20) | (0.24) |
| Time * damage (multiple) | −0.03*** | −0.45* | −0.60** |
| | (0.01) | (0.19) | (0.23) |
| Constant | 3.86*** | 80.19*** | 77.78*** |
| | (0.03) | (0.67) | (0.60) |
| Observations | 10256 | 9100 | 9686 |
| Log likelihood | −10 104.58 | −36 205.01 | −38 020.51 |
| Akaike Inf. Crit. | 20 239.16 | 72 440.02 | 76 071.02 |
| Bayesian Inf. Crit. | 20 347.69 | 72 546.76 | 76 178.69 |

*P<0.05; **p<0.01; ***p<0.001.
†Stress-related health symptoms were reverse-coded such that higher levels indicate less stress.
PGA, peak ground acceleration.

To highlight the implications of the findings, we distinguished poor and good health on the basis of health scores, enabling us to compute ORs and relative risk. For mental health, we used the conventional criterion of MHI <60 as cut-off.[31] For perceived health, we classified 'good' and 'outstanding' as good health and all other scale points as poor (conform international convention). For symptoms, we devised our own cut-off based on distributional characteristics combined with content criteria: a classification of <60 as poor health resulted in 9% of the unaffected population being classified as such. ORs

were calculated in weighted models, controlling for age, education and gender.

## Public involvement

The research setup (design and outcome measures) was discussed with an advisory board consisting of institutions (eg, local municipalities) and representatives of the public (eg, action groups). The present work has been disseminated in a public report.

## RESULTS
### Sample characteristics

There were no significant fluctuations in sample composition over time in terms of gender, education level and damage to own housing (see table 2). Young respondents were under-represented. There was attrition during the study. Dropout characteristics revealed no differences between exposed versus control groups and no association between dropout and health. Analyses showed no indications that attrition influenced any of the effects reported below. Over time, the average age of participants increased, as young people tended to have a higher likelihood of dropout. It is important to note that additional analyses found no significant interaction effect between age and exposure, suggesting that the effects of exposure were age-independent. Because the sample was not entirely representative and attrition relatively high, we carefully checked the potential consequences thereof and found no indications this influenced results.

Regarding levels of exposure, we know from the damage claims register (Provided by the institution handling damage claims, the Centrum voor Veilig Wonen) that the rates of exposure vary substantially within the region: in central areas up to 100% of homes have reported damage at least once. Outside these areas, there is progressively less damage. A substantial part of the province has (nearly) no damage. Average levels of damage are closely associated with ground motion:[25] in postal-code areas where 0% damage was reported until January 2016, there was hardly any exposure to ground motion (total ground motion $PGA_{cum} = 0.07\,mm/s^2$). Only 3% of the sample located in this area suspected having damage due to earthquakes. In the areas where up to 20% damage was previously reported, total ground motion was somewhat higher $PGA_{cum} = 0.64\,mm/s^2$) and more people, 26% of the sample, indicated suspecting they have damage. And in the areas where 20% to 100% had reported damage, ground motion was considerably higher, $PGA_{cum} = 4.13\,mm/s^2$ and a high percentage of our sample, 83%, suspected having damage.

### The impact of exposure to gas extraction on health over time

The analyses of conditional growth models on self-rated health, stress-related health problems and mental health showed consistent results across all three indicators. Table 3 shows the final results for all variables.

Important to note is that, after including control variables in step 1, there was a significant effect of exposure to physical ground motion ($PGA_{cum}$) on all three health indicators: more ground motion was associated with poorer health. The effect of time was also significant: over time, health deteriorated.

In step 2, we included damage to housing. Having damage once had no significant effect on any of the health indicators. Only participants with multiple damages experienced negative health consequences.

The effects of ground motion were suppressed by the larger effects of exposure to multiple damage on all health indicators (p's<0.01). The suppression occurs because damage and total ground motion are strongly correlated. It does not mean that the association between exposure to ground motion and health should be disregarded: there might, for example, be some health effects of 'peak exposure' to strong ground motion in the weeks or months after an earthquake. The current analysis does not address such peak exposure effects because it only assesses average impact on health over the entire 2-year period and gradual changes in health over time.

In step 3, the significant 'multiple damage (vs no damage) X time' interaction reveals that exposure to multiple damages is associated with a deterioration of health over time. The inclusion of this interaction variable improved model fit.

To interpret these effects of exposure to damage and assess their magnitude, we calculated ORs for health measures at every time point, as well as the average impact of exposure over time (table 4). Inhabitants exposed to damage once are only marginally (and not significantly) affected compared with a no-damage control group (averaged ORs range from 1.10 to 1.20). Those exposed to damage multiple times are more likely to report poor self-rated health (OR=1.64, with a 95% CI of 1.31;2.04), more stress-related health symptoms (OR=2.52 (1.89 to 3.38)) and less good mental health (OR=1.83 (1.40 to 2.39)) than those without damage. This indicates that damage has a considerable impact on participants' health (we also investigated whether women's health is affected differently by this stressor than men's, but as evidenced in online supplemental table S2, this is not the case).

The table also suggests that differences between groups increase over time. ORs for the difference between those with multiple damage and no damage is considerably higher 21 months after first measurement for self-rated health (OR=2.00 (1.57 to 2.55)), stress related health symptoms (OR=3.36 (2.45 to 4.68)) and mental health (OR=2.38 (1.78 to 3.21)). In terms of relative risk, this means that those whose homes have multiple damage at T5 are 1.60 (1.37 to 1.86) times more likely to report poor health, 2.84 (2.14 to 3.76) times more at risk of elevated levels of stress related health symptoms and 2.11 (1.63 to 2.74) times more likely to report negative mental health.

We also compared the weighted means of the ORs of control variables that are known correlates of health (age, gender, level of education) to the effect of damage

**Table 4** Proportion of participants who have poor health and OR of participants who have poor health with damage (compared with those with no damage) across measurements, with 95% CIs—adjusted for age, gender and level of education (Netherlands 2016–2017)

| Measurement | Damage | Percentage poor health | OR |
|---|---|---|---|
| *Self-rated health* | | | |
| T1 | None | 22.2% (19.9% to 24.5%) | – |
| | One time | 22.5% (19.6% to 25.4%) | 1.02 (0.82 to 1.26) |
| | Multiple | 25.6% (22.7% to 28.4%) | 1.21 (0.99 to 1.47) |
| T3 | None | 21.6% (18.8% to 24.3%) | – |
| | One time | 24.4% (20.5% to 28.3%) | 1.17 (0.90 to 1.53) |
| | Multiple | 32.4% (29.2% to 35.7%) | 1.75 (1.41 to 2.18) |
| T4 | None | 21.3% (18.4% to 24.2%) | – |
| | One time | 30.0% (25.7% to 34.4%) | 1.60 (1.22 to 2.09) |
| | Multiple | 35.5% (31.8% to 39.2%) | 2.06 (1.63 to 2.61) |
| T5 | None | 23.6% (20.3% to 26.9%) | – |
| | One time | 27.5% (22.9% to 32.1%) | 1.23 (0.92 to 1.65) |
| | Multiple | 38.0% (34.0% to 42.0%) | 2.00 (1.57 to 2.55) |
| Weighted average | None | 22.1% (19.4% to 24.9%) | – |
| | One time | 25.6% (21.8% to 29.4%) | 1.20 (0.93 to 1.55) |
| | Multiple | 31.8% (28.5% to 35.2%) | 1.64 (1.31 to 2.04) |
| *Symptoms* | | | |
| T1 | None | 9.2% (7.7% to 10.8%) | – |
| | One time | 10.0% (7.9% to 12.1%) | 1.09 (0.81 to 1.47) |
| | Multiple | 17.3% (14.9% to 19.7%) | 2.08 (1.62 to 2.68) |
| T3 | None | 7.1% (5.5% to 8.8%) | – |
| | One time | 6.5% (4.3% to 8.6%) | 0.90 (0.58 to 1.37) |
| | Multiple | 13.7% (11.4% to 16.1%) | 2.09 (1.55 to 2.85) |
| T4 | None | 8.1% (6.2% to 10.0%) | – |
| | One time | 9.4% (6.7% to 12.1%) | 1.18 (0.78 to 1.75) |
| | Multiple | 21.8% (18.7% to 25.0%) | 3.24 (2.40 to 4.42) |
| T5 | None | 7.1% (5.3% to 9.0%) | – |
| | One time | 9.1% (6.3% to 11.9%) | 1.30 (0.84 to 1.99) |
| | Multiple | 20.3% (17.0% to 23.5%) | 3.36 (2.45 to 4.68) |
| Weighted average | None | 8.0% (6.4% to 9.8%) | – |
| | One time | 8.8% (6.5% to 11.2%) | 1.10 (0.75 to 1.60) |
| | Multiple | 18.0% (15.3% to 20.7%) | 2.52 (1.89 to 3.38) |
| *Mental health* | | | |
| T1 | None | 8.5% (7.0% to 10.0%) | – |
| | One time | 9.0% (7.0% to 10.9%) | 1.06 (0.78 to 1.43) |
| | Multiple | 12.4% (10.3% to 14.5%) | 1.53 (1.18 to 1.98) |
| T2 | None | 8.5% (6.8% to 10.2%) | – |
| | One time | 9.2% (6.9% to 11.6%) | 1.09 (0.77 to 1.54) |
| | Multiple | 18.1% (15.6% to 20.7%) | 2.40 (1.86 to 3.13) |
| T3 | None | 11.1% (9.0% to 13.2%) | – |
| | One time | 12.0% (9.2% to 14.9%) | 1.10 (0.79 to 1.51) |
| | Multiple | 14.5% (12.1% to 16.9%) | 1.36 (1.04 to 1.77) |

**Table 4** Continued

| Measurement | Damage | Percentage poor health | OR |
|---|---|---|---|
| T4 | None | 11.9% (9.6% to 14.1%) | – |
| | One time | 11.8% (8.9% to 14.7%) | 0.99 (0.71 to 1.38) |
| | Multiple | 20.3% (17.2% to 23.4%) | 1.9 (1.46 to 2.47) |
| T5 | None | 9.0% (6.9% to 11.1%) | – |
| | One time | 12.5% (9.3% to 15.7%) | 1.44 (0.99 to 2.07) |
| | Multiple | 19.1% (15.9% to 22.2%) | 2.38 (1.78 to 3.21) |
| Weighted average | None | 9.7% (7.8% to 11.5%) | – |
| | One time | 10.6% (8.1% to 13.2%) | 1.11 (0.80 to 1.55) |
| | Multiple | 16.4% (13.8% to 19.0%) | 1.83 (1.40 to 2.39) |

Scores were categorised as low health as follows: (1) very poor, poor, or fair perceived health; (2) a score below 60 for stress related health symptoms and (3) a score below 60 for mental health.

(figure 1). Across the three health measures, effect sizes (ORs) of damage are slightly larger than those of education (for a high vs low level of education, the average ORs over time are 0.53 (0.41 to 0.68) for self-rated health; OR=0.56 (0.41 to 0.79) for stress-related health symptoms and OR=0.58 (0.42 to 0.81) for mental health).

## DISCUSSION

Natural and induced seismicity can have negative consequences for local populations due to (acute or accumulated) health threats and irreversible changes to the living environment. Yet, so far studies have not assessed the accumulated impact of (the consequences of) induced seismicity on (psychosomatic) health *over time*. Moreover, most studies lack a non-exposure control group. The present study aimed to address these shortcomings by studying the impact of exposure to gas extraction (and subsequent damage to housing), compared with a no-exposure control group, on health over a time period of 21 months. Our study provides strong indications that exposure to negative side-effects of induced seismicity (eg, damage to people's homes) constitutes an increasing health risk over time: we found that those who self-reported having multiple damages to housing

experienced more negative health consequences than those without damage. Moreover, these effects increased over time. Results showed that chronic physical exposure to ground motion (assessed objectively) was also related to health, although less strongly than reporting multiple personal damages.

To our knowledge, this is the only study of the long-term impact of induced seismicity on health. Therefore, we can only compare our results with the long-term impacts of very different types of disaster - limiting comparability. For one, the Chernobyl nuclear disaster: study participants lived in a seriously contaminated area approximately 50 miles from Chernobyl. 6.5 years postdisaster, inhabitants were twice as likely to have negative self-rated health (OR: 2.25 (1.96 to 2.58)) and psychological distress (OR: 1.93 (1.69 to 2.22)), compared with a non-exposed control group.[32] Chernobyl clearly constitutes a very different type of disaster and health risk (radiation exposure). The Brisbane floods were also very different in many respects (eg, sudden disaster onset; deaths) but with some comparable outcomes, such as considerable damage to homes. 6–7 months postdisaster, those exposed to flooding were twice as likely to report psychological distress compared with the non-exposed.[33] It appears that the health impact

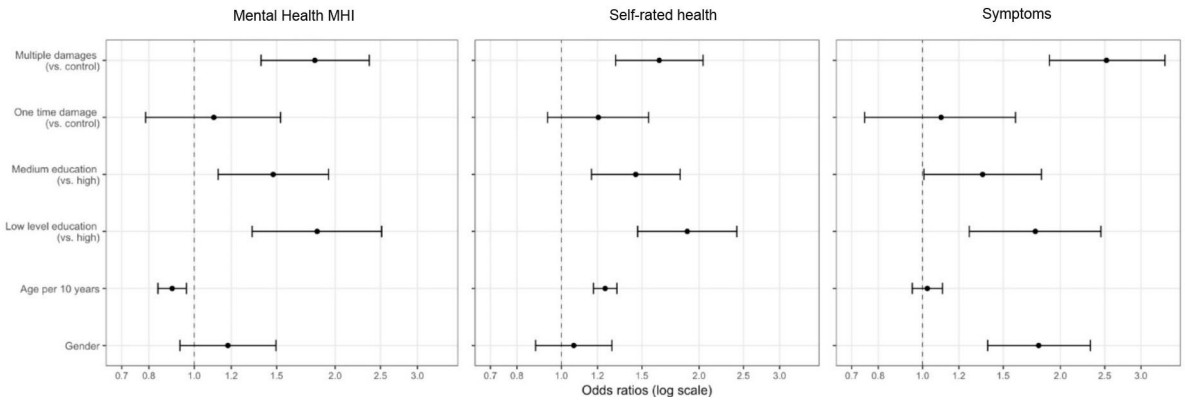

**Figure 1** Weighted average ORs. MHI, Mental Health Inventory.

of these very different and in many ways more 'acute' disasters are, in terms of effect size, somewhat comparable to the health impact of the more chronic exposure to induced earthquakes caused by gas extraction. One potential reason for the comparable effect sizes is that our study focused not just on the environmental effect (eg, amount of total ground motion) but zoomed in on the subgroup who were severely affected because they had multiple instances of damage to their own home. We further speculate that the man-made nature of the hazard (the fact that earthquakes are induced) may also enhance the impact on the population.

The present work also provides first time insights into the development of (psychosomatic) health symptoms in response to chronic disaster. In the area of acute disaster response, studies on longitudinal health impact reveal that distress decreases over time,[34 35] implying recovery of victims. Yet looking at discussions comparing chronic man-made (technological) disasters to acute natural disasters, we see the present context shares elements identified as reasons for potential long-term health impact of technological disasters: a strong element of culpability in causing disaster concerns about damage compensation after disaster and uncertainty regarding when disaster impact will end ('the book is never closed'; p. 148).[12 36–38] In line with this work, our findings suggest that for chronic disasters/hazards, negative effects can accumulate over time, presumably because the recurrent threat and poor crisis response leads to an accumulation of stress.

### Limitations

A potential limitation of this sample could be concerns about its representativeness: for one, attrition was 45.3% over time and younger respondents were somewhat under-represented. However, attrition was no different for the exposed and non-exposed groups, was unrelated to health outcomes and all further analyses suggest that neither attrition nor sample characteristics had any substantial influence on results and conclusions drawn. Second, there might be an influence of confounding variables. Yet we believe effect sizes are robust: (1) the exposed and control groups were very similar regarding key population and geographical characteristics; (2) follow-up analyses revealed no interactions between any of the population characteristics and the effects of exposure.

Third, responses could have been biased because participants knew the survey was about the social impact of gas extraction. It is relevant here that an 'objective' exposure measure (PGA) revealed comparable health outcomes to self-reported exposure. Moreover, analyses on a cross-sectional representative sample of residents (n=16 340) in the 2016 health monitor of Statistics Netherlands, the National Institute for Public Health and public health services found comparable results. In this survey, the study intent was not clear.

One of the three health measures included, stress-related health symptoms, was an adaptation of a previously validated symptoms list,[39] shortened for this specific study. Although the shortened version was not previously validated, it was psychometrically sound. Also, patterns are comparable across health measures, two of which are validated.

One of our exposure measures is self-reported damage. It is possible that damage is perceived differently depending on people's health status. Importantly, physical exposure to ground motion was associated with significant health effects. But effects of damage were stronger. This could be because damage is a more precise and proximate indicator of how individuals are affected by exposure and also because of recursive effects of (mental) health on perceived damage.

An important issue is generalisability: is the situation in Groningen comparable to other areas with induced seismicity (eg, fracking, wastewater injections)? We can only make reasoned inferences. Induced earthquakes are relatively common in energy projects which involve injection.[40] A priori, similar health consequences could occur in all sites in which populations are affected by induced earthquakes. Moreover, the vulnerability of people exposed to seismicity is likely influenced by similar factors: negative consequences are man-made and involve safety, health and social risks.[12 13] In sum, although more research on the impact of induced seismicity is needed,[41] we suggest effects are likely to generalise beyond the Groningen case.

### Practical implications

The consequences of induced seismicity pose challenges to decision-makers. Benefits to the public good need to be balanced against the welfare of local populations.[21] As projects involving induced seismicity rapidly grow, governments and businesses face decisions whether to invest and how to manage risks. Our work provides a case study of what occurs if seismicity is not kept in check. It can increase awareness of the vulnerability of exposed populations and provide important input for future decision making, monitoring and contingency planning.

### Conclusion

Recent years have seen a rise in induced seismicity. Little is known about the (longitudinal) impact thereof on (psychosomatic) health. The present study is the first to our knowledge evidencing the long-term impact of induced seismicity on health.

**Contributors** KS, BK, JR, JB, FO, FG and TP contributed to the research questions and study design. JR and FO contributed to data collection. BK and TP conducted statistical analyses. KS, BK and TP interpreted the results and wrote the initial draft of the manuscript. All authors commented on the final draft of the manuscript. BK and TP had full access to the data in the study and can take responsibility for the integrity of the data and the accuracy of the data analysis.

**Funding** This study was funded by the National Coordinator Groningen. Award/grant number: not applicable.

**Competing interests** None declared.

**Patient consent for publication** Not required.

**Ethics approval** All procedures performed in studies involving human participants were in accordance with the ethical standards of the ethical board of the Department of Psychology of the University of Groningen, The Netherlands (research code ppo-015-085).

**Provenance and peer review** Not commissioned; externally peer reviewed.

**Data availability statement** Data are not available in order to guarantee anonymity of participants.

**ORCID iD**
Katherine Stroebe http://orcid.org/0000-0002-3933-1531

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
