## [Reviewer comments · BMJ Open]

ARTICLE DETAILS

TITLE (PROVISIONAL)	Chronic disaster impact: the long-term psychological and physical health consequences of housing damage due to induced earthquakes
AUTHORS	Stroebe, Katherine; Kanis, Babet; Richardson, Justin; Oldersma, Frans; Broer, Jan; Greven, Frans; Postmes, Tom

VERSION 1 – REVIEW

REVIEWER	Sae Ochi Jikei University School of Medicine, Japan
REVIEW RETURNED	16-Jun-2020

GENERAL COMMENTS	I think the authors well responded to my comments. I am sorry for overlooking the authors' description about ethical consideration.
---

REVIEWER	Ben Beaglehole Department of Psychological Medicine University of Otago New Zealand
REVIEW RETURNED	27-Jul-2020

GENERAL COMMENTS	Hi this is my first time reviewing the paper which appears to have undergone peer review and extensive revision previously. I have no major comments or concerns. The study design is appropriate to answer the question and the analysis and discussion are clear. Three other points follow: 1. A paper authored by myself provides further support to the introduction, and chosen methodology and may be helpful https://www.cambridge.org/core/journals/the-british-journal-of-psychiatry/article/psychological-distress-and-psychiatric-disorder-after-natural-disasters-systematic-review-and-metaanalysis/D84B03CEC50473E56938D2C09CD7464E2. The sentence in the results beginning "Looking at the ORs, you see..." is overly casual, does not report findings, and should be rephrased.3. The limitations section should acknowledge the response rate more explicitly and be included in the discussion on response bias
--

REVIEWER	Eizaburo Tanaka Hyogo Institute for Traumatic Stress
REVIEW RETURNED	09-Aug-2020

GENERAL COMMENTS	Review comment
----------------

This study deals with the long-term psychosomatic health issues related to induced earthquakes by gas extraction among the residents in Groningen. The authors compared the exposed population with the non-exposed one in terms of perceived health, stress-related health symptoms, and mental health with the 21 months follow-up cohort. The manuscript reports that the exposed population showed significant adverse outcomes of both psychological and physical health. Multiple damages to the resident houses were associated with worse health during the 21 months follow-up period. Overall, the authors have made a good attempt at adding value to the discussion of the long-term health impact of the man-made low-intensity chronic disaster. Besides, they made a massive effort to address the previous reviewers' comments, and it could deserve to be published. However, it still needs some clarifications and minor revisions.

For the introduction,

#1 Please add information on the context of the Groningen gas field for one paragraph.

The Authors described the context of the Groningen gas field in response to the reviewer's comment (p10-11). It would be helpful to understand this study context for readers if the summary was provided in the introduction. In particular, Fig 2 the authors presented is informative.

#2 In the second paragraph, the authors stated "Naturally occurring seismicity is associated with mental health problems in survivors (e.g., depression, PTSD)[3,4]. These studies are generally cross-sectional and lack an unexposed control group [3]." The second sentence is misleading because there are a lot of cohort studies for survivors of natural disasters, including earthquake.

For the methods,

#3 Please provide the response rate at baseline in the text. Is it 18% (N=4556/25000) or 86% (N=3937/4556)? How many residents were asked to participate in this survey? And of all invited, how many agreed to response?

#4 How did you define education level? Please state it in the "Study Variables" of methods clearly.

#5 The instruments of health outcomes No.1(The WHO and Statistics...) and No.3(MHI-5) seem to be validated in the language used here. If so, please state it in the "Study Variables" of methods clearly.

#6 When did the authors inquire about the "personal exposure to damage due to gas extraction"?

Was it just at the baseline (T1) or multiple times (T1 to T5)? Does the variable "Damage to house" in Table 2 mean "personal exposure to damage due to gas extraction" at the baseline (T1)? Since this is the exposure variable, it is better to be defined clearly in the text.

#7 Did the authors excluded the participants who showed poor health based on health score they defined from the baseline population? As is well known, the temporality (the effect has to occur after the cause) is crucial to understand the causal relationship. If not, please conduct additional analysis to use the baseline population without poor health.

	For the discussion, #8 I understand there are not many similar previous studies to be compared for the discussion. Therefore, I encourage the authors to provide their hypothesis of why and how the low-intensity chronic man-made earthquake can cause health problems in the discussion based on the available evidence. For the abstract, #9 In the conclusion, the authors stated that “It identifies which subpopulation is particularly at risk and why”. However, there is no information regarding this statement in the results of the abstract. The abstract should be standalone, and its context must be consistent.
--	---

REVIEWER	Guang-Ming Han Department of Dermatology and Rheumatology, Dermatology Hospital of Southern Medical University, Guangzhou, China.
REVIEW RETURNED	23-Oct-2020

GENERAL COMMENTS	1. Statistical methods and analyses: in the manuscript, according to the design, the study belongs to a longitudinal survey. Longitudinal studies, in which repeated measures are obtained over time from each subject, are one important and commonly used type of repeated measures study. Therefore, the authors need regard the repeated measurements as a cluster of correlated measures within each individual, the correlations arising from the shared individual characteristics. In general, a generalized estimating equation (GEE) is a popular technique for the analysis of repeated measurements data. Therefore, the authors need to analyze the data with GEE instead of the model with interaction between damage and time. 2. Missing data: there are many lost for participants from T2 to T5. “Total number of participants participating in that measurement, decline of number of participants participating as compared to the number of participants participating at T1 (19.8% for T2; 33.1% for T3; 40.2% for T4 and 45.3% for T5)” in Table 2. Therefore, the significant different results for damage and/or time may contribute to the large missing data. Therefore, the authors need to compare and provide the characteristics between the participants who stay in the study and the participants who lost from the study.
--

VERSION 1 – AUTHOR RESPONSE

Reviewer: 1

Reviewer Name: Sae Ochi

Institution and Country: Jikei University School of Medicine, Japan

Competing interests: None declared

Comments to the Author

1. I think the authors well responded to my comments. I am sorry for overlooking the authors' description about ethical consideration.

Response: We thank Reviewer 1 for this positive feedback on our manuscript.

Reviewer: 2

Reviewer Name: Ben Beaglehole

Institution and Country:

Department of Psychological Medicine

University of Otago

New Zealand

Competing interests: None declared

Comments to the Author

Hi

this is my first time reviewing the paper which appears to have undergone peer review and extensive revision previously.

I have no major comments or concerns. The study design is appropriate to answer the question and the analysis and discussion are clear. Three other points follow:

1. A paper authored by myself provides further support to the introduction, and chosen methodology and may be helpful

<https://www.cambridge.org/core/journals/the-british-journal-of-psychiatry/article/psychological-distress-and-psychiatric-disorder-after-natural-disasters-systematic-review-and-metaanalysis/D84B03CEC50473E56938D2C09CD7464E>

Response: Thank you for this reference. It is indeed very relevant, also in stressing the importance of longitudinal designs and control groups in disaster studies. We have integrated it into the introduction.

2. The sentence in the results beginning "Looking at the ORs, you see..." is overly casual, does not report findings, and should be rephrased.

Response: *We have rephrased this paragraph and included odds ratios with confidence intervals for level of education.*

3. The limitations section should acknowledge the response rate more explicitly and be included in the discussion on response bias

Response: *If we understand correctly, Reviewer 2 asks us to include the actual response rate more explicitly both in the limitations and discussion sections. In both sections we have now added levels of attrition. In addition we have integrated the attrition and response bias section of the discussion to improve readability.*

Reviewer: 3

Reviewer Name: Eizaburo Tanaka

Institution and Country: Hyogo Institute for Traumatic Stress

Competing interests: None

Comments to the Author

Review comment

This study deals with the long-term psychosomatic health issues related to induced earthquakes by gas extraction among the residents in Groningen. The authors compared the exposed population with the non-exposed one in terms of perceived health, stress-related health symptoms, and mental health with the 21 months follow-up cohort. The manuscript reports that the exposed population showed significant adverse outcomes of both psychological and physical health. Multiple damages to the resident' houses were associated with worse health during the 21 months follow-up period. Overall, the authors have made a good attempt at adding value to the discussion of the long-term health impact of the man-made low-intensity chronic disaster. Besides, they made a massive effort to address the previous reviewers' comments, and it could deserve to be published. However, it still needs some clarifications and minor revisions.

Response: We thank Reviewer 3 for this positive feedback about our manuscript - and the confidence in its publishability.

For the introduction,

#1 Please add information on the context of the Groningen gas field for one paragraph.

The Authors described the context of the Groningen gas field in response to the reviewer's comment (p10-11). It would be helpful to understand this study context for readers if the summary was provided in the introduction. In particular, Fig 2 the authors presented is informative.

Response: Reviewer 3 asks us to provide more information about the 'Groningen context' in the introduction. We agree it helps to clarify the study context at an earlier stage and have now added this

information at the end of the introduction when we introduce the research question. We also refer to (new) supplementary material here, in which we provide additional information regarding seismicity in this province (including former Fig. 2).

#2 In the second paragraph, the authors stated “Naturally occurring seismicity is associated with mental health problems in survivors (e.g., depression, PTSD)[3,4]. These studies are generally cross-sectional and lack an unexposed control group [3].” The second sentence is misleading because there are a lot of cohort studies for survivors of natural disasters, including earthquake.

Response: Reviewer 3 points out that our sentence regarding cross-sectionality is misleading. We agree there has been an increase in longitudinal studies of seismicity in recent years. At the same time, various researchers have observed that cross sectionality and lack of control group are a shortcoming for the majority of studies in the area of natural disasters. We have clarified the sources we rely on in making this assessment and we have adapted this sentence as follows: While there has been some increase in studies considering longitudinal health effects of seismicity, lack of longitudinal design and an unexposed control group have been highlighted as a concern in studies on natural disasters (Beaglehole et al., 2018; Dai et al., 2016; Goldmann & Galea, 2014).

For the methods,

#3 Please provide the response rate at baseline in the text. Is it 18% (N=4556/25000) or 86% (N=3937/4556)? How many residents were asked to participate in this survey? And of all invited, how many agreed to response?

Response: *In the sample and recruitment section we now clearly describe response rates to the initial invitation to sign up for the study and the response rate to the questionnaire: “Eighteen percent (N=4577) signed up for the study, and later received invitations to all questionnaires. Of these 4577, 86% (3943) filled out the first questionnaire.”*

#4 How did you define education level? Please state it in the “Study Variables” of methods clearly.

Response: *We now provide additional information regarding levels of education. The categorization of (Dutch) levels of education was as follows: ‘low’ (no, elementary, or pre-vocational education), ‘middle’ (secondary or vocational education), or ‘high’ (higher education level). The classification we used is based on Statistics Netherlands (CBS).*

#5 The instruments of health outcomes No.1(The WHO and Statistics...) and No.3(MHI-5) seem to be validated in the language used here. If so, please state it in the “Study Variables” of methods clearly. -

Response: We provide detailed information about validation now. See also our response to associate editor above.

#6 When did the authors inquire about the “personal exposure to damage due to gas extraction”?

Was it just at the baseline (T1) or multiple times (T1 to T5)? Does the variable “Damage to house” in Table 2 mean “personal exposure to damage due to gas extraction” at the baseline (T1)? Since this is the exposure variable, it is better to be defined clearly in the text.

Response: We inquired about this at every measure. The text has been clarified also in the table. We define this more clearly in the method section too.

#7 Did the authors excluded the participants who showed poor health based on health score they defined from the baseline population? As is well known, the temporality (the effect has to occur after the cause) is crucial to understand the causal relationship. If not, please conduct additional analysis to use the baseline population without poor health.

Response: *The reviewer correctly points out that those who already have poor health scores might not be the most diagnostic of change over time. We have now conducted additional analyses in which we model the same conditional growth models as already presented in the paper but only with respondents whom we defined as ‘healthy’ on T1 for each health measure individually. We compared these models to those presented in the paper. Both the pattern of results and the magnitudes of the effects (and therefore the sizes of p-values) we find are very similar to the results presented in the paper. Therefore we decided to not include these additional analyses in the paper. The reviewer suggests that we remove participants with poor health from the sample. Because the models with and without that group are essentially the same, we decided to retain the entire sample in the models we present in the paper. This, after all, provides the best and most complete overview of health change in the population as a whole.*

For the discussion,

#8 I understand there are not many similar previous studies to be compared for the discussion. Therefore, I encourage the authors to provide their hypothesis of why and how the low-intensity chronic man-made earthquake can cause health problems in the discussion based on the available evidence.

Response: We thank the reviewer for this interesting suggestion. We have now included the following discussion regarding the long term impact of chronic technological disasters: “Yet looking at discussions comparing chronic man-made (technological) disasters to acute natural disasters, we see reason to expect long term health impacts of the gas extraction: This is because the gas extraction context contains elements similar to those provided as reasons for potential long-term health impact of technological disasters: A strong element of culpability in causing disaster, concerns about damage compensation after disaster and uncertainty regarding when disaster impact will end (“the book is never closed”; p.148, Erikson, 2010; Baum, 1993; Couch & Kroll-Smith, 1985; Picou, 2000; Picou et al., 2004). Our findings suggest that for chronic disasters/hazards, negative effects can accumulate over time, presumably because the recurrent threat leads to an accumulation of stress.”

For the abstract,

#9 In the conclusion, the authors stated that “It identifies which subpopulation is particularly at risk and why”. However, there is no information regarding this statement in the results of the abstract. The abstract should be standalone, and its context must be consistent. -

Response: Thank you for attending us to this point. We have adapted this sentence to identify the subpopulation (those with repeated damages to housing).

Reviewer: 4

Reviewer Name: Guang-Ming Han

Institution and Country: Department of Dermatology and Rheumatology, Dermatology Hospital of Southern Medical University, Guangzhou, China.

Competing interests: None declared

Comments to the Author

1. Statistical methods and analyses: in the manuscript, according to the design, the study belongs to a longitudinal survey. Longitudinal studies, in which repeated measures are obtained over time from each subject, are one important and commonly used type of repeated measures study. Therefore, the authors need regard the repeated measurements as a cluster of correlated measures within each individual, the correlations arising from the shared individual characteristics. In general, a generalized estimating equation (GEE) is a popular technique for the analysis of repeated measurements data. Therefore, the authors need to analyze the data with GEE instead of the model with interaction between damage and time.

Response: The reviewer is correct. The caption of Table 3 contained an error, which was misleading. It read “Unstandardized regression parameter estimates”. We apologize for this. Reviewer correctly points out that a linear regression would not have been appropriate for data such as these. Indeed we did not conduct a regression but a conditional growth model. This was stated in the text but due to the faulty table caption this was unclear. We changed the caption to: “Results of multilevel conditional growth models: Unstandardized parameter estimates” to clarify that we did not conduct regression analyses but growth models, which do indeed take into account the dependencies within individuals and which are specifically designed to model change over time within individuals.

Reviewer recommends that we use GEE. This is an alternative but it is not one we think is as suitable in this case. GEE has a number of strengths, but conditional growth models have specific advantages (esp. the treatment of missing values and its suitability for datasets with a limited number of repeated measures) which mean that we have a strong preference to use CGM in this case. Due to its handling of missingness, the GEE models can be fitted only to data completed all five time points (N=1480) and would therefore be less robust. And CGM is a good method, suitable for the present purposes of assessing fixed effects of damage.

2. Missing data: there are many lost for participants from T2 to T5. “Total number of participants participating in that measurement, decline of number of participants participating as compared to the number of participants participating at T1 (19.9% for T2; 32.9% for T3; 40.2% for T4 and 45.3% for T5)” in Table 2. Therefore, the significant different results for damage and/or time may contribute to the large missing data. Therefore, the authors need to compare and provide the characteristics between the participants who stay in the study and the participants who lost from the study.

Response: *We conducted additional analyses to speak to this question. First, we conducted the same conditional growth analyses as presented in the paper, but only with participants who filled in the questionnaire at t5 (N=2150). We compared these models to those presented in the paper. The pattern of results we find are very similar to the results presented in the paper. Second, we did the same thing, but only with participants who filled out all five questionnaires (N=1480). Again, effects show the same patterns as presented in the paper, though we do lose some power.*

Lastly, we compared age, gender, level of education, and damage to house for participants who only filled in the first questionnaire (N=538) versus participants who filled in the first and the last questionnaire (N=2072). The results are displayed in this table:

Var	Participants who left after T1	Participants who filled in at least T1 and T5
Total N	538	2072
Age (mean)	50.46	60.00
Low education	123 (23%)	538 (26%)
Mid education	181 (34%)	633 (31%)
High education	150 (28%)	851 (41%)
Male	240 (45%)	1066 (51%)
Female	223 (41%)	987 (48%)

No damage	186 (35%)	804 (39%)
One time damage	98 (18%)	493 (24%)
Repeated damage	133 (25%)	558 (27%)

The participants who discontinued are considerably younger and have on average lower education levels (well known attrition effects). However, attrition does not appear to be related to damage.

VERSION 2 – REVIEW

REVIEWER	Ben Beaglehole Department of Psychological Medicine University of Otago New Zealand
REVIEW RETURNED	07-Jan-2021

GENERAL COMMENTS	This paper has obviously been reviewed on a number of occasions including by myself. The topic is of interest and the results are important. On re-reading the article I think issues relating to exposure and outcome could be tightened further. 1. The bullet point strengths and limitations section state that the study is examining the health consequences of "gas extraction". I don't think this is precise enough. The study seeks to study the mental health consequences of manmade earthquakes caused by gas extraction. 2. I think the issue of exposure/non-exposure is important. Perhaps this area could be improved. As I understand it, you have used two measures of exposure. PGA and housing damage. PGA is associated with poor mental health but only if housing damage occurred. I infer from this that damaging people's homes in a setting of mistrust/poor compensation is the key factor in causing poor mental health. And that repeated shaking is not the issue. I am unsure if the use of two measures of exposure is the best approach statistically and whether or not regarding earthquake damage as a mediating factor is better although I don't think this will affect the substance of your paper. I would leave this for you to consider possibly with the input of a statistician.
---

REVIEWER	Eizaburo Tanaka Hyogo Institute for Traumatic Stress
REVIEW RETURNED	04-Feb-2021

GENERAL COMMENTS	I think the authors addressed my concerns appropriately. Now it's worth publication.
--

VERSION 2 – AUTHOR RESPONSE

Reviewer: 2

Dr. Ben Beaglehole, University of Otago

Comments to the Author:

This paper has obviously been reviewed on a number of occasions including by myself. The topic is of interest and the results are important.

Response: We thank the reviewer for this positive feedback.

On re-reading the article I think issues relating to exposure and outcome could be tightened further.

1. The bullet point strengths and limitations section state that the study is examining the health consequences of "gas extraction". I don't think this is precise enough. The study seeks to study the mental health consequences of manmade earthquakes caused by gas extraction.

Response: We have made the suggested changes (referring to (psychosomatic) rather than mental health in order to be consistent with the abstract).

2. I think the issue of exposure/non-exposure is important. Perhaps this area could be improved. As I understand it, you have used two measures of exposure. PGA and housing damage. PGA is associated with poor mental health but only if housing damage occurred. I infer from this that damaging people's homes in a setting of mistrust/poor compensation is the key factor in causing poor mental health. And that repeated shaking is not the issue. I am unsure if the use of two measures of exposure is the best approach statistically and whether or not regarding earthquake damage as a mediating factor is better although I don't think this will affect the substance of your paper. I would leave this for you to consider possibly with the input of a statistician.

Response: Many thanks for this observation. We realised that we needed to clarify the issue because the previous version of the paper was a bit ambiguous. Both measures of exposure, individually, have a significant relationship with health outcomes. Thus, PGA is associated with poor health and damage is, too. These two factors are highly correlated, but of the two damage has a somewhat stronger effect. There are multiple plausible explanations for this, the most likely being that damage is a very direct and consequential personal experience which adds to any aversive effects of experiencing an earthquake. But we can not be sure of this: all we have are the correlations and strengths of association between the two exposure measures and health. And the correlations being what they are, if both predictors are entered into the same model together, the effects of damage suppress those of PGA.

Whilst this suppression does confirm that damage is the stronger predictor in this set of analyses, it does not support the inference that repeated shaking isn't an issue. As we now hope to have clarified in the paper: Cumulative PGA is an environmental indicator that adds together all major earthquakes that together could have been traumatic (an objective indicator that adds up the total amount of ground motion on a particular location). Damage is a self-report of whether a person believes to have suffered damage to their home that is due to these earthquakes. Because of these differences in the nature of the exposure variables, we can not easily draw inferences of the kind you suggest from the empirical evidence that objective effects of cumulative PGA are suppressed by self-reported damage.

In order to draw the inferences that you would like us to reflect on, we would really need to study closely how people respond to ground motion and that is not what the current dataset and statistical model allow us to do. Specifically, to make inferences about ground motion and its health effects, one could conduct a longitudinal study in which self-reports of exposure to ground motion as well as self-reports of subsequent damage are included (and ideally objective measures of both too). We would then need to make assessments of these variables plus health at more time points (and crucially to include measures after a few larger earthquakes too) and use another statistical model.

We have reason to believe that exposure to ground motion, in and of itself, can be traumatic too, especially in the aftermath of a serious earthquake during which people are acutely reminded of the fragility of their home and the uncontrollability of these events (and therefore even in the absence of damage). But in the longer run, the acute trauma may subside somewhat, whereas the long-term nuisances and stress for all those who suffer damage are known to be considerable. And as mentioned the risk of damage is simply much higher in high cumulative PGA areas. In other words, the kind of suppression effect that we have demonstrated here is a consequence of the kind of data we have collected and the (long-term) analysis we conducted. We have tried to clarify this in the text by making explicit what can and can not be concluded from this. We thank the reviewer for this comment: it is important to be precise and to clarify that statistical suppression should not lead one to disregard the impact that ground motion on its own could have.

Reviewer: 3

Dr. Eizaburo Tanaka, Hyogo Institute for Traumatic Stress

Comments to the Author:

I think the authors addressed my concerns appropriately. Now it's worth publication.

Response: We thank the reviewer for this positive feedback.

VERSION 3 – REVIEW

REVIEWER	Ben Beaglehole Department of Psychological Medicine University of Otago New Zealand
REVIEW RETURNED	08-Apr-2021

GENERAL COMMENTS	Thanks for your response to my latest comments
--